# Architecture Engineering and Construction Industrial Framework for Circular Economy: Development of a Circular Construction Site Methodology

**Clelia Fagone [1], Margherita Santamicone [2] and Valentina Villa [1,\*]**

1   Department of Structural, Geotechnical and Building Engineering, Politecnico di Torino, 10129 Turin, Italy
2   Environment, Safety Environment and Quality, Webuild S.p.A., 00156 Rome, Italy
\*   Correspondence: valentina.villa@polito.it

**Abstract:** All sectors have been affected by digital, economic, and demographic transitions. These will also be completely changing the paradigms of the construction industry, which is increasingly being pressured to implement strategic solutions in order to mitigate its own current environmental impact. Existing environmental standards and voluntary protocols are useful tools for reducing the environmental impact of buildings and infrastructures above all during the operation phase, but there is no consistent methodology for creating a circular and sustainable construction stage at the moment, probably because this represents a small stage in the entire life cycle of a structure; nevertheless, the construction industry impacts a lot on the environment and should be managed in a circular way. A detailed analysis of the current problem is provided, as well as a potential solution involving a transition to a circular economic model for the entire construction lifecycle, supported by innovative models and methodologies. The purpose of this article is to create a "Circular Construction Sites" (CCS) flow using circular concepts and to develop a circular methodology for construction sustainability, considering practical solutions related to materials, water, and waste management. According to the circular economy model, the circular construction site can communicate with other industrial ecosystems to promote external circularity and zero waste production. A set of potential key performance indicators (KPIs) for monitoring circularity processes is presented. Finally, a method is proposed that circularizes not only the building or infrastructure design, but also the construction phase.

**Keywords:** circular economy; construction management; waste management

## 1. Introduction

It was 2009 when the Ellen McArthur Foundation was founded in the UK, with the aim of 'Let us build a circular economy', defining the circular economy as 'a systems solution framework that tackles global challenges like climate change, biodiversity loss, waste, and pollution' [1]. It was in 2014 that the topic became truly international, when the European Commission presented the circular economy package 'The Missing Link: A European Action Plan for the Circular Economy' [2].

The circular economy thus became a strategic objective and is a new economic model on which much research has now been carried out [1–11].

The first article on the circular economy dates back to 2007, and more than two thirds of the total 101 publications listed on the term are from 2015–2017 [12].

At the international level, global stakeholders such as the OECD1, WEF1, and UNEP1 have also expressed the urgency of finding efficient technologies and methodologies to close material cycles by issuing several reports on the circular economy [12–17]. Japan and China were the first to do so, having formally introduced green policies at the national level. In Europe, following the transposition of the circular economy directive, Denmark, Germany, The Netherlands, and the United Kingdom are activating policies and programs

to implement this transition [2,18]. The aim of the circular economy is to develop new economic models that can reverse current wasteful trends and have a positive impact on society and the environment, in terms of reducing the use of natural resources by generating virtuous social, environmental, and economic impact [19].

Regarding the construction sector and the built environment, this involves designing infrastructures and buildings that minimize the extraction of new raw materials, favoring the use of recycled products, and thinking about the end-of-life reuse of materials and components [20]. An interesting concept expressed by Niero, M. and Hauschild, M.Z., 2017 [21], describes circularity as "maintaining products, components and materials at their maximum usefulness at all times".

As this article shows, large differences manifest themselves globally with regard to the CE, yet the potential ascribed to the CE of breaking the global "take, make, consume and dispose" namely to create a pattern of growth—a linear model based on the assumption that "resources are abundant, available, easy to source and cheap to dispose of ( . . . )" [22]—is widely shared among different societal actors across the globe.

Therefore, it becomes imperative to look for an economic model that can address these critical issues of unsustainability in the system. The concept of the circular economy is certainly the first response to these issues and is considered a valid alternative in order to produce and consume more responsibly, triggering virtuous processes of the recycling and reuse of raw materials. The CE represents a sustainable growth strategy because it allows the 'decoupling' of resource use from economic growth, thus contributing to sustainable development [3,5,15,22,23].

This new strategic approach contains numerous concepts such as the reuse and/or recycling and remanufacturing of safe materials that are safe for people and the environment according to the Cradle-to-Cradle Material Health quality categories [24]. In this direction, i.e., in reducing the environmental impact of the built environment, are also all the efforts spent on integrating the use of renewable energies, such as photovoltaics and solar panels, and on reducing the waste of drinking water.

A characteristic element of the circular economy is the valorization of the positive economic, environmental, and social impacts [10,11,21], which allow the sustainability assessment of projects and constructions to be completed in a systemic way [5]. However, the circular economy is a very complex research topic, with extensive theoretical approaches and practical tools that can change dynamics, whether they are political, social, or economic [9]. There are several European Union reports on environmental impact and natural resource use [2,18] of sectors which highlight the high intensity of use of resources and the production of waste pertaining to construction.

The Architecture, Engineering, Construction and Operation (AECO) sector, according to a European Union directive [25] and also reported in a Spanish study [26], has an impact on the EU economic sector of approximately 40% of gross final energy consumption; 35% of greenhouse gas emissions; 50% of material resources extracted; 30% of water consumption; and 35% of waste generation. Therefore, switching to a CE paradigm in the built environment would have a major impact in terms of reducing pressure on non-renewable resources and waste generated.

Introducing circular economy concepts to the construction sector also means significantly limiting the environmental impact of human activities related to buildings and infrastructure and helps reduce the impact of the built environment throughout its life cycle, creating resilient buildings that can respond more effectively to environmental, energy, and economic crises [27]. Despite the urgency of having a sustainable economic system and despite the huge impact of the Architecture Engineering and Construction (AEC) sector, a comprehensive methodology for monitoring and evaluating circularity in the sector has not yet been formulated. In recent years, some benchmark indices for calculating and monitoring the environmental impact of construction have been introduced, such as the Building Circularity BC Index [27], which assesses materials and their demountability, or the proposed framework for buildings built according to circular economy principles, which

incorporates some new indicators into the Building Research Establishment's (BREEAM) environmental assessment methodology [28], or the work of the International Organization for Standardization (ISO) which focusses on implementing new circular metrics through ISO/TC 323. Furthermore, there are already structured methods such as the Cradle to Cradle (C2C) certification for designing buildings with safe and certified products for the circular economy [29].

The purpose of this paper is to create a methodology for assessing the degree of compliance with the aforementioned CE principles by developing a clear and unambiguous framework for the holistic measurement and evaluation of circularity. The developed methodology, in turn, could become a sound basis for the development of new policies and provide guidance to building designers on the impact of adopting certain circular economy measures on the circularity of a new building or major renovation activity.

In the first part of this paper, an overview of the main tools and frameworks available in the market to approach the construction sector in a sustainable and innovative way is presented. Then, a potential methodology to manage a "Circular Construction Site" is presented, by referring to the new circular economy model which is spreading. Starting from existing European standards and national decrees about construction materials, water, and energy use and management, and also considering the voluntary environmental protocols that are used to award a building, a series of steps that should be adopted to improve the impact of a construction site is defined. The methodology is based on a general construction site process (Section 3) that was mapped covering the stages that usually characterize each type of construction site. Specific attention is given to the choice, use, and management of materials (Section 3.1), water (Section 3.2), machineries (Section 3.3), and waste (Section 3.4) on-site. Therefore, starting from this general process, it is possible for each specific project to dress up the methodology and enrich the steps by focusing on it. Parameters and key performance indicators (KPIs) are proposed (Section 5), on the basis of existing indicators, to monitor and manage the methodology during the construction phase and to have direct feedback about the methodology.

## 2. Sustainability in AEC Industry

On 28 July 2022, we had already exhausted our planet bio-capacity. The Global Footprint Network measures the ecological footprint and calculates the Overshoot Day each year; it declares that "humans use as much ecological resources as if we lived on 1.75 Earths" [30]. People have migrated from villages and the countryside to cities and urban areas; urbanization has increased deforestation and habitat loss and decreased biodiversity; and more than 8 billion people now populate the planet, "from the beginning of time on Earth to the start of the 20th century, the population of the planet grew from zero to 1.6 billion". Then, thanks to many factors, "the population increased to 6.1 billion in just 100 years, which is an almost fourfold increase in the number of humans over a relatively short period"; this creates an overpopulation that needs more resources and increases greenhouse gases [31].

It is essential to achieve sustainable solutions from an environmental, social, and economic point of view. The design, build, use, and maintenance of construction work has a big influence in terms of sustainable cities, use of resources, and climate impacts. Some solutions have been already implemented mainly in the design and use phases to promote sustainability and take care of the environment. The construction stage has never been discussed in detail and no methodology approach has already been developed to manage a sustainable site. The European Union produces more than 2.5 tons of waste every year and the highest percentage is from the construction sector [32]. The main problem is related to the fact that we do not have unlimited quantities of resources. In the last few years, it has been trying to adopt a new economy model that moves from a linear to a circular model to face today's challenges. The Ellen Macarthur Foundation defined the circular economy (CE) as "tackling global challenges like climate change, biodiversity loss, waste, and pollution.

It is based on three principles, driven by design: eliminate waste and pollution, circulate products and materials (at their highest value), and regenerate nature" [33].

There are different ways to adopt the CE, but the idea behind it is to create a circular production process that allows one to use existing materials for as long as possible, as shown in Figure 1. The AEC lifecycle is broken down into four stages with small suggestions that should be adopted to facilitate the transition. In the design phase, there are some strategies that have been created. It is essential, in order to move to the circular model, to predispose the design to disassembly. The construction phase, nowadays, undergoes the choices made in the previous phases. The LCA of a product allows one to evaluate the environmental footprint along its entire life cycle in an analytic and systematic way.

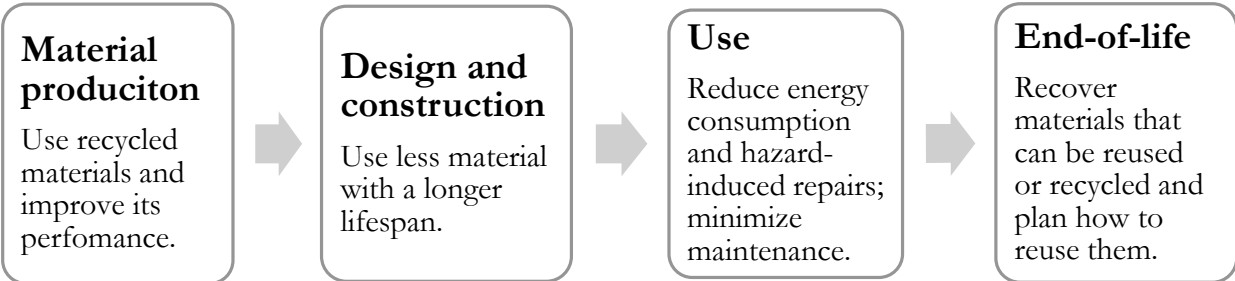

**Figure 1.** The process for improving AEC lifecycle.

The CE model aims to achieve the cradle-to-cradle strategy, including reuse or recycling operations, instead of finishing the LCA in the end-of-life stage. Due to all of the world's challenges, nowadays it is becoming increasingly common to attribute to each good factor a voluntary label that assesses its properties and characteristics. The European standard EN 15804:2012+A2:2019 [34] suggests an integration of EPD in the construction sector. Type III environmental declarations (EPDs) are based on the LCA of a product conducted according to rules and prerequisites. It is mandatory to deliver an LCA paper that includes the following modules: A1–A3 (production process and energy and material consumption), C1–C4 (demolition/deconstruction process, transport, disposal, and recycling of the product) and D (Figure 2).

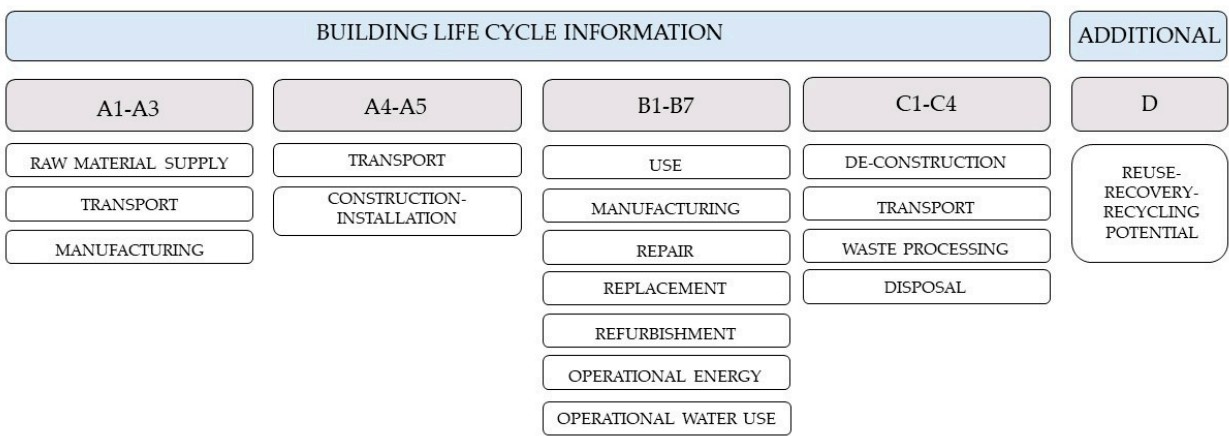

**Figure 2.** Building lifecycle following the EN 15978:2011 [34] indications.

Since the 1990s, several environmental protocols have been created to raise awareness and spread a culture aimed at building sustainability. They are voluntary and country-oriented but aim to ensure designers' responsibility towards the environment. The main goal of these protocols is to give a ranking to a building and certify it from an environmental point of view. Some parameters could also be adopted in the construction stage to assess and manage a circular construction site.

To understand which are the main topics and characteristics of these protocols in this paragraph, they are compared in Table 1 which also highlights the points regarding strengths and weaknesses. It is useful to verify which protocols, and in which terms, take into account the A-stages, product, and construction phases, through indicators or criteria that have to be checked to carry out the work in a more sustainable way.

**Table 1.** Comparison of different environmental protocols in the construction industry.

| | Release Year | Country | Certification Level | Main Categories | Strengths and Weaknesses |
|---|---|---|---|---|---|
| **LEED [35]** | 1998 | USA | * Platinum<br>* Gold<br>* Silver<br>* Certified | - Integrative process<br>- Location and transportation<br>- Sustainable sites<br>- Water efficiency<br>- Energy and atmosphere<br>- Materials and re-sources<br>- Indoor environ-mental quality<br>- Innovation<br>- Regional priority | It is the most widely used even if it is based upon US systems; it requires a huge quantity of documents but usually data are easily available; and mixed building functions are difficult to assess. It takes into account the innovation sector and has an index to evaluate the integrative process as a whole. |
| **BREEAM [36]** | 1990 | UK | * Outstanding<br>* Excellent<br>* Very good<br>* Good<br>* Pass<br>* Unclassified | - Management<br>- Health and wellbeing<br>- Energy<br>- Transport<br>- Water<br>- Material<br>- Waste<br>- Land use and ecology<br>- Pollution<br>- Innovation | It is the first protocol developed; it allows comparisons and benchmarking of different buildings; it is independently audited but it follows very exact requirements; and the weighted system is complex. For that reason, it can result in limited openness and transparency. Even this protocol considers the innovative solutions. |
| **ITACA [37]** | 2004 | Italy | A global evaluation given by the average value of all of the parameters. (Between −1 and 5) | - Quality of site<br>- Resource consumption<br>- Environmental load<br>- Indoor environmental quality<br>- Quality of the service | It is limited to the Italian market because it does not have an English version; and it has a complex system. Economic and social aspects are not well developed; instead, environmental load and resource consumption are very detailed. It does not present a certification value but a global average value. Its results are region oriented. |
| **DGNB [38]** | 2009 | Germany | * Platinum<br>* Gold<br>* Silver<br>* Bronze | - Environmental quality<br>- Economic quality<br>- Sociocultural and functional quality<br>- Technical quality<br>- Process quality<br>- Site quality | It has a clear structure still in development; and it presents assistance from an auditor. It is available in the English version, which allows for the expansion of the market, even if the high competition in this market usually alienates new users. |
| **HQE [39]** | 2004 | France | * Excellent<br>* Very good<br>* Good<br>* Pass | - Energy<br>- Environment<br>- Health<br>- Comfort | Even though the English version was recently launched it is not used so much outside of France. |
| **CASBEE [40]** | 2001 | Japan | * Excellent<br>* Very good<br>* Good<br>* Fairly poor<br>* Poor | - Indoor environment<br>- Quality of service<br>- Outdoor environment on-site<br>- Energy<br>- Resource and materials<br>- Off-site environment | It has a rating method that is different from all the other protocols analyzed and presents several charts and outputs. It is based on Japanese standards, so its results are hard to adopt in other countries. |

**Table 1.** *Cont.*

| | Release Year | Country | Certification Level | Main Categories | Strengths and Weaknesses |
|---|---|---|---|---|---|
| **GREEN STAR [41]** | 2003 | Australia | * One star<br>* Two stars<br>* Three stars<br>* Four stars<br>* Five stars<br>* Six stars | - Management<br>- Indoor environment quality<br>- Energy<br>- Transport<br>- Water<br>- Materials<br>- Land use and ecology<br>- Emissions<br>- Innovation | It is based on LEED and BREEAM protocols. It is more suitable for hot climate countries. LEED and BREEAM also consider the impact of innovation solutions. |

A direct consequence that these protocols consider with regard to the entire life cycle of a building is that most of these parameters refer to the use phase, mainly in terms of energy but also in terms of waste and water. This is not a limit; the analysis was conducted just to have a general view of what is taken into account and in which way. Some parameters could also be adopted to the construction stage to assess and manage a circular construction site. Starting from these data, it is possible to analyze the construction phase and all of the choices that can be made to consider the construction of a structure in a circular way.

## 3. Construction Site Process

In order to create and describe a solid methodology that governs the circular process of a construction site, it is necessary to have a broad but clear understanding of all construction practices in order to decide whether to create circularity within the construction site itself or within the external environment. Even though construction represents a tiny portion of a structure's life cycle, it may have a significant influence on the environment. It must be emphasized that there are several sorts of construction sites based on their dimensions, soil conditions, and building types. In general, it is feasible to differentiate between temporary and movable construction sites where building or civil engineering work is performed; they are distinguished by a fenced-in area including plants, machinery, storage facilities, and temporary structures. After the completion of the project, the construction site is dismantled and, in certain instances, relocated and utilized for the execution of a new project.

The flowchart depicts a typical construction site procedure for a new building. The outcome is shown schematically in Figure 3: a succession of activities (rectangles), interspersed with inquiries (rhombuses), and final tests or actions to validate the conclusion of a process (the gray shapes). In the chart, it is possible to highlight broad solutions that serve as inputs to produce circularity; these solutions, which will be discussed and explained in separate paragraphs, are highlighted in light-gray rectangles with a bold font.

To simplify, the first phase of a construction site includes the delineation and examination of the site's area, based on what was stated during the design phase. Before beginning the job, it may be required to perform remediation so that personnel may reside in uncontaminated areas. Terrains may include hazardous chemicals and pollutants that must be eliminated from the location. When destruction is necessary, it may be worthwhile to evaluate whether there are items that may be recovered, reused, or recycled on-site or elsewhere; if not, it is necessary to indicate where to dispose of garbage that must be sorted on-site.

When the site is clear and uncontaminated, the execution of works may begin. To begin, it is essential to understand the necessary building procedures and the list of supplies. The executive project already specifies the materials that must be utilized, but in this phase it is possible to choose the supply in order to create sustainable and circular choices, as indicated in the section on materials below (Section 3.1. Materials). They may be repurposed by-products from the same facility or outside materials. Once the material supply has been identified, it must be kept and classified in a designated space, with special care for hazardous items (e.g., those that are flammable).

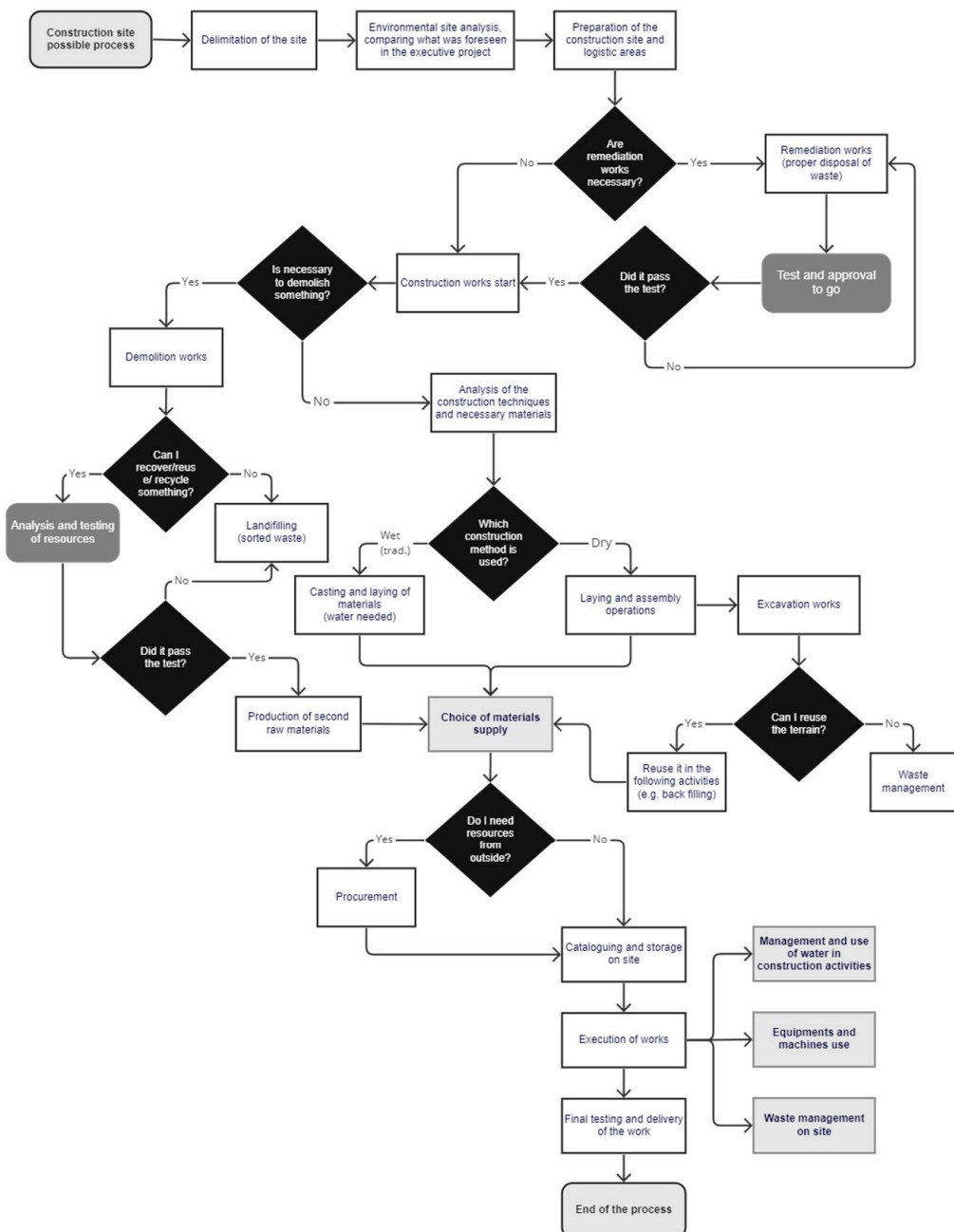

**Figure 3.** Construction site process flowchart.

The traditional construction techniques necessitate the use of water since they involve the casting and placing of materials such as concrete on-site. Specific remarks on the usage of water in construction activities are offered in the next section (Section 3.2. Water). Dry structures are developing solutions comprising prefabricated objects or modular components that must be erected without the use of water and may be removed with ease. Excavation work performed above the groundwater table is also termed a dry activity. If the soil is devoid of hazardous materials, it may be utilized for operations such as backfilling; otherwise, it is disposed of in an authorized landfill.

The execution of works is performed through the use of specific equipment and machineries that are powered and could require energy. Since the use of energy has an important role in improving the performance of the site, that part of the chart is discussed in detail in a specific section (Section 3.3. Plant, Machineries, and Equipment). During each of these phases, waste production is expected; for this reason, waste management is needed

which allows one to predispose logistic areas to store sorted waste. A deeper discussion is presented below (Section 3.4. Waste). Once the work is finished, a finale test is carried out and delivered.

As stated before, the design of this flowchart emphasizes the existence of four significant areas that need in-depth analysis: materials, water, machineries and equipment, and waste. A circular construction site is defined by the usage and management of these four components. As a result, the following paragraphs offer additional flowchart components to enhance the basic flowchart by including potential information relating to these four categories. To encourage circularity in construction sites, it is essential to emphasize the breadth of these concerns, which are examined only from an environmental sustainability standpoint.

### 3.1. Materials

Construction materials play a central role in the construction of a structure; they are selected during the design phase based on the type and performance of the structure being built and the construction procedures being used. Furthermore, they rely upon the nation in which the project is being constructed and the availability of raw resources there.

During the construction phase, it is possible to choose the material supply and make circular considerations in addition to those already chosen during the design phase. As is represented in Figure 4, material supply is connected with the waste management on-site and with the reuse of demolition waste, if demolition works are planned.

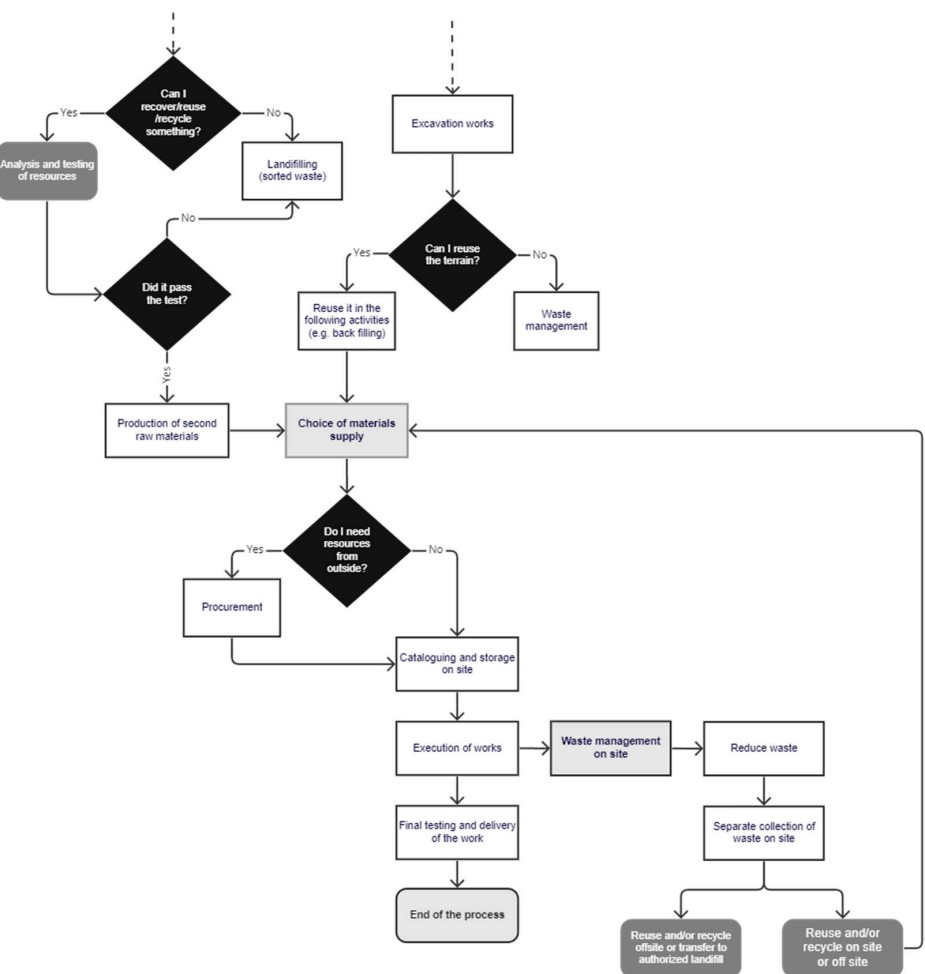

**Figure 4.** Flow chart of choice of material supply.

Typically, following proper testing, some on-site resources may be recovered, reused, or recycled to generate by-products or secondary raw materials both on-site and off-site.

Several options exist to increase sustainability and promote the CE model in the construction phase, including the reuse of existing structures; recovering/recycling materials; the use of local materials; the disassembly of materials; and the selection of certified materials.

CAM (Criteri Minimi Ambientali, or Minimum Environmental Criteria) in Italy is a set of environmental standards specified for the whole lifespan of a product. "They became effective with article 18 of Italian Law 221/2015 [42] and article 34 of Legislative Decree 50/2016 "Procurement Code" [43] (amended by Legislative Decree 56/2017 [44]) carrying "Energy and environmental sustainability standards"".

The 11 October 2017 Ministerial Decree [45] allowed the use of CAM for 18 categories, one of which being construction focused. The following characteristics are similar to all construction components [46]:

- Disassemblability: at least fifty percent by weight of building components, excluding installations, must be recyclable or reused at the end of their life, and at least fifteen percent of this proportion must comprise non-structural elements.
- Material recovered or recycled: The percentage of recovered or recycled material in the building's materials must be at least 15% based on the overall weight of the components. At least 5% of this proportion must consist of non-structural elements. If present in the building materials analyzed in Section 2.4.2, this proportion must be evaluated.
- Hazardous substances.

According to the law, the designer must produce documentation and certificates to verify these standards.

To promote sustainability and the CE on a construction site, the selection of externally sourced materials should adhere to a set of conditions that provide monitoring of the manufacturing stage. A solution might be to adopt a type of material passport that identifies the material's qualities and essential information over the product's full LCA.

In the European project BAMB (Buildings as Material Banks) [47], an electronic materials passport was developed to:

- Increase the value or maintain the value of materials, products, and components over time;
- Create incentives for suppliers to produce healthy, sustainable, and circular materials/building products;
- Support material choices in reversible building design projects;
- Make it easier for developers, managers, and renovators to choose healthy, sustainable, and circular materials/building products.

This enables the consideration of building materials in a circular loop that disregards waste production and the end-of-life phase. All of the information is included on a single sheet of paper to facilitate the transmission of information between participants in the building chain. The BAMB proposal depicts this paper as a solution to be made on-site, but in order to enhance the CE throughout the building phase it should be prepared by the material manufacturers and then, if required, modified on-site.

### 3.2. Water

Water is primarily good as its presence in the construction site is fundamental for workers, construction activities, and the washing of machines and equipment. An efficient management and use of water on-site can improve circularity and avoid waste and wastefulness. As is shown in Figure 5, water, preferably non-potable water, is used in construction activities. It is important to know its physical, chemical, and biological properties in order to avoid the overcoming of certain legal [48] and operational limits (process use feasibility), for example color, pH-value, and different contaminants such as metal, chloride, and salts.

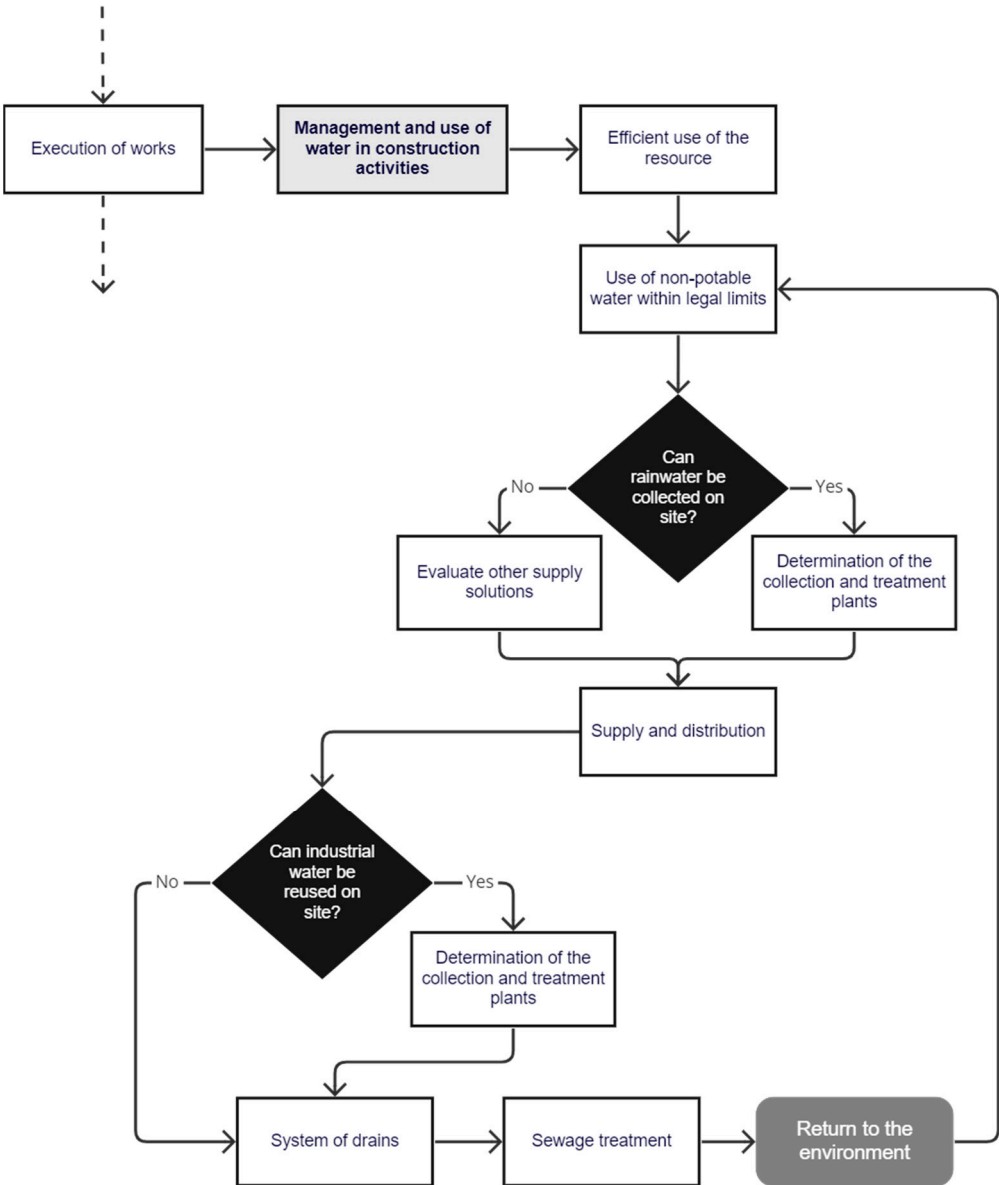

**Figure 5.** Flowchart of the water use and management in construction sites.

Promoting a method that enables water to be collected in specific harvesting systems, treated, and then reused or returned to the environment is a sustainable and creative approach. Different types of collecting systems may be found in a construction site, e.g., rainfall and industrial wastewater. Depending on the scale of the construction site, it may be feasible to install a system that collects, treats, and stores non-potable rainwater. As stated in the Italian Legislative Decree n° 152/2006 article n° 113 [49], rainfall typically does not need any special treatment if not potentially contaminated. The only exceptions are runoff rainwater brought in via separate pipes and first flush water or washing water from outdoor locations. Rainwater collecting requires surfaces and channels that route the water to a collection system. There, numerous processes enable the water to be filtered and purified for storage in tanks and reuse. Sometimes, water quantity coming from rainwater collecting systems cannot satisfy the demands of a construction site both in terms of quantity and continuity, depending on the site's geographical and climatic conditions. Furthermore, there is often a lack of enough sensitivity to consider water as a valuable resource, and the low cost of withdrawal makes creating water recovery and recycling facility systems not always cost-effective.

Any sort of industrial effluent must be treated before release or be reused in other industrial processes. A cost–benefit analysis has to be implemented in order to verify whether to install a water treatment plant that ensures discharge permit compliance (in compliance with leg. Decree 152/06 III Part) [49] or manage the wastewater as waste (in compliance with leg. Decree 152/06 IV Part) [49].

### 3.3. Plant, Machineries, and Equipment

On a construction site, machinery and equipment are constantly present in accordance with the scheduled work activities. They may remain on-site for the duration of construction or for a limited time only. Large construction businesses may own their own equipment, but in recent years it has become increasingly popular for enterprises to rent them; this is the ideal option from a CE standpoint. Even in this instance, sustainable observations might be conducted in order to highlight how the machinery and equipment are fueled. The primary distinction between this flowchart and the others described in this paper is the absence of circularity. In actuality, owing to our emphasis on power, we must address energy and fuel supplies. When considering energy from an environmental perspective, it is feasible to adopt sustainable practices, such as the usage of energy from renewable sources; however, this cannot be called a circular process since power does not return to its source. Nevertheless, it is worth mentioning and considering for future research.

Before describing the flowchart and beginning to explain the sustainable factors taken into account, it is vital to identify the primary construction equipment. They may be divided into four categories depending on their function and application:

- Earth-moving equipment: excavators, graders, loaders, skid loaders, crawler loaders, backhoes, bulldozers, trenchers, scrapers, wheeled loading shovels;
- Construction vehicles: tippers, dumpers, trailers, tankers;
- Material-handling equipment: cranes, conveyors, hoists, forklifts;
- Construction equipment: concrete mixture, compactors, pavers, road rollers.

Depending on the kind of engine, they may be either electrical- or gas-powered, and hybrid alternatives have recently emerged. The majority of material-handling equipment is electrical, while all other vehicles are diesel-powered. Some car manufacturers have previously introduced electric or hybrid solutions that are powered by batteries, but since these solutions are smaller and less powerful than conventional ones, they have not been generally embraced.

Returning to the flowchart shown in Figure 6, if a piece of equipment is fuel-powered and no other alternatives are available, it is required that its emissions are quantified and a compensatory solution is created. An alternative to a circular economy model is a green economy model, which is "described as low carbon, resource efficient, and socially inclusive" [50]. Respecting an environment with limited resources, the sustainable model as a whole tries to live in harmony with the environment. This notion enables the adoption of solutions such as the usage of biofuel, a fuel derived from biomass such as plant, algae, or animal waste.

It is permissible to discover a technique to compensate for $CO_2$ emissions via reforestation or afforestation, the protection of parks or natural reserves, soil management, ocean fertilization, or all of the solutions that function as the lungs of the Earth, absorbing carbon dioxide and exhaling oxygen.

When referring to electrical equipment, it is conceivable that energy is generated on-site or that an external energy supply is selected. In both circumstances, power should be derived from renewable sources. It is feasible to install on-site solar panels, wind turbines, and other technologies that generate energy from renewable sources. If the energy supply is external, it is recommended to check the guarantee of origin (GO) or the supplier's renewable energy certificate (REC) to ensure that it comes from renewable sources.

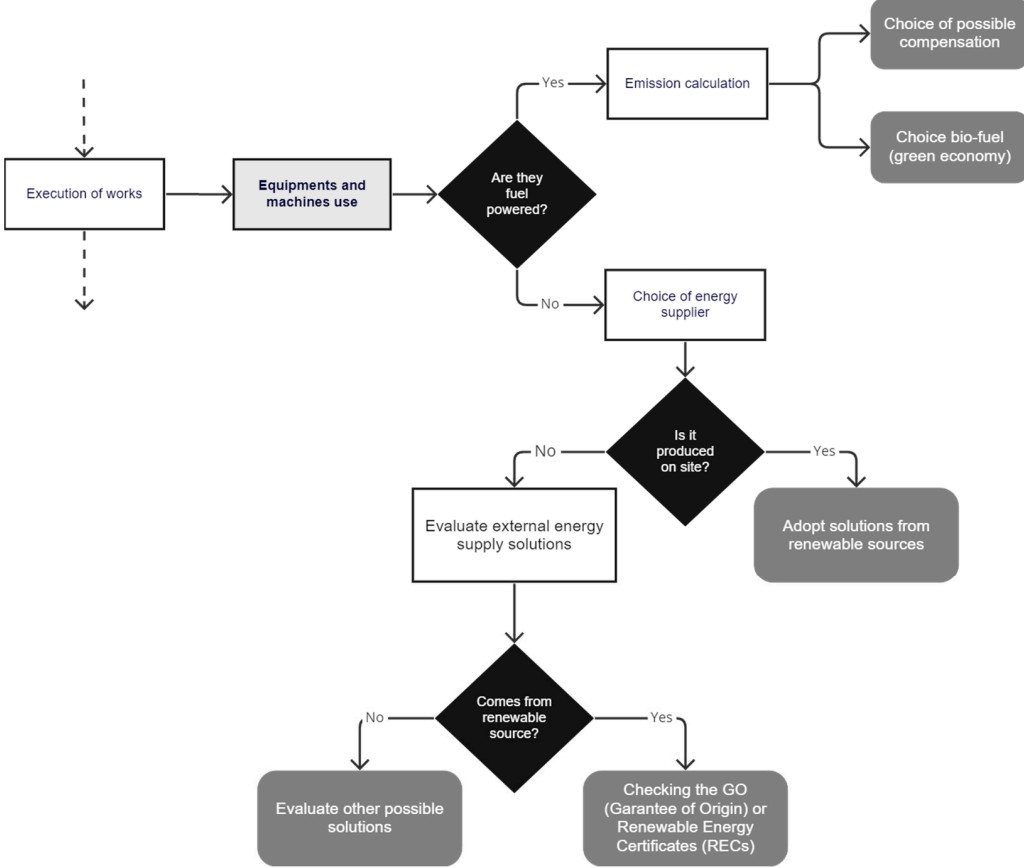

**Figure 6.** Flowchart of construction machineries and equipment feeding.

*3.4. Waste*

During the execution of works, wastes are produced on-site and it is important to predispose an area for where they are collected and sorted to decide their future destination. Waste can come from demolition works or construction activities; we refer to it as construction and demolition (C&D) waste. Presented in the flowchart (Figure 7) are the circular process proposed to foresee an efficient use of resources in order to reduce the amount of waste produced on-site and a plan to recover, reuse, and recycle them on-site or off-site.

In 2016, the European Union commission published the "EU Construction and Demolition Waste Protocol" [51] to regulate waste management in construction sites and promote the recycling of materials. The protocol suggests to:

(a) Improve waste identification, source separation, and collection;
(b) Improve waste logistics;
(c) Improve waste processing;
(d) Improve quality management;
(e) Appropriate policy and framework conditions.

All of these actions can be implemented on-site for better waste management and easier resource revaluing. The protocol refers to the C&D waste list taken from the "European Waste Catalogue" (EWC) [52].

While the objective of the waste management system is to efficiently plan the waste flow outside of the construction site by avoiding its disposal in landfills, it is important to pay attention to the legal requirements concerning the end-of-waste (EoW) procedures.

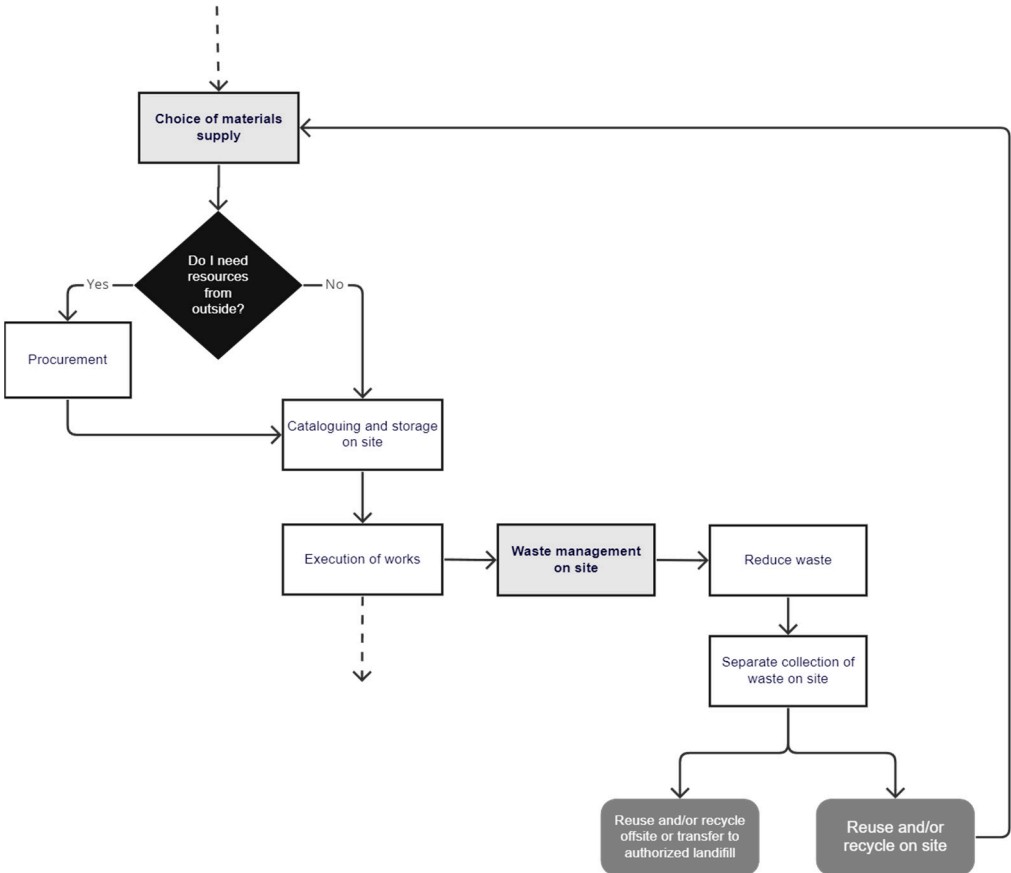

**Figure 7.** Flowchart of waste management for a construction site.

### 4. The Circular Construction Site Methodology

Using the circular economy concepts and all of the considerations made in the previous chapter, it is possible to develop the so called "Circular Construction Site" (CCS) methodology. Methodologies often present extremely abstract topics; for this reason, it was helpful to try to represent these concepts concretely within the construction site ecosystem (Figure 8). The result aims to understand how construction site activities can communicate with each other in a circular way and how the construction site itself can communicate with the external environment. The ideal situation promoted by the CE model is a mutual exchange between circular industrial ecosystems, even between totally different sectors to improve the product's value chain (e.g., agricultural waste such as straw and seaweed used in green buildings). Construction sites are only one type of industrial ecosystem and deserve an in-depth analysis to explain their internal circularity.

A circular construction site must be characterized by a system that allows one to manage and collect scrap materials and waste in each process and predispose their reuse. The construction process can be synthetized through three steps:

- Material supply and storage: any type of product or system arrives on-site and is stored in a specific area; there, the packing is removed and it is ready to be used;
- Work on-site: the product can be laid on-site or may have to undergo a process; usually, a processing area is planned. This is the core of the process and requires the use of specific machines and equipment and sometimes even water;
- Laying: this last phase allows the structure to be built by laying the final product in the right place; equipment is also needed at this stage.

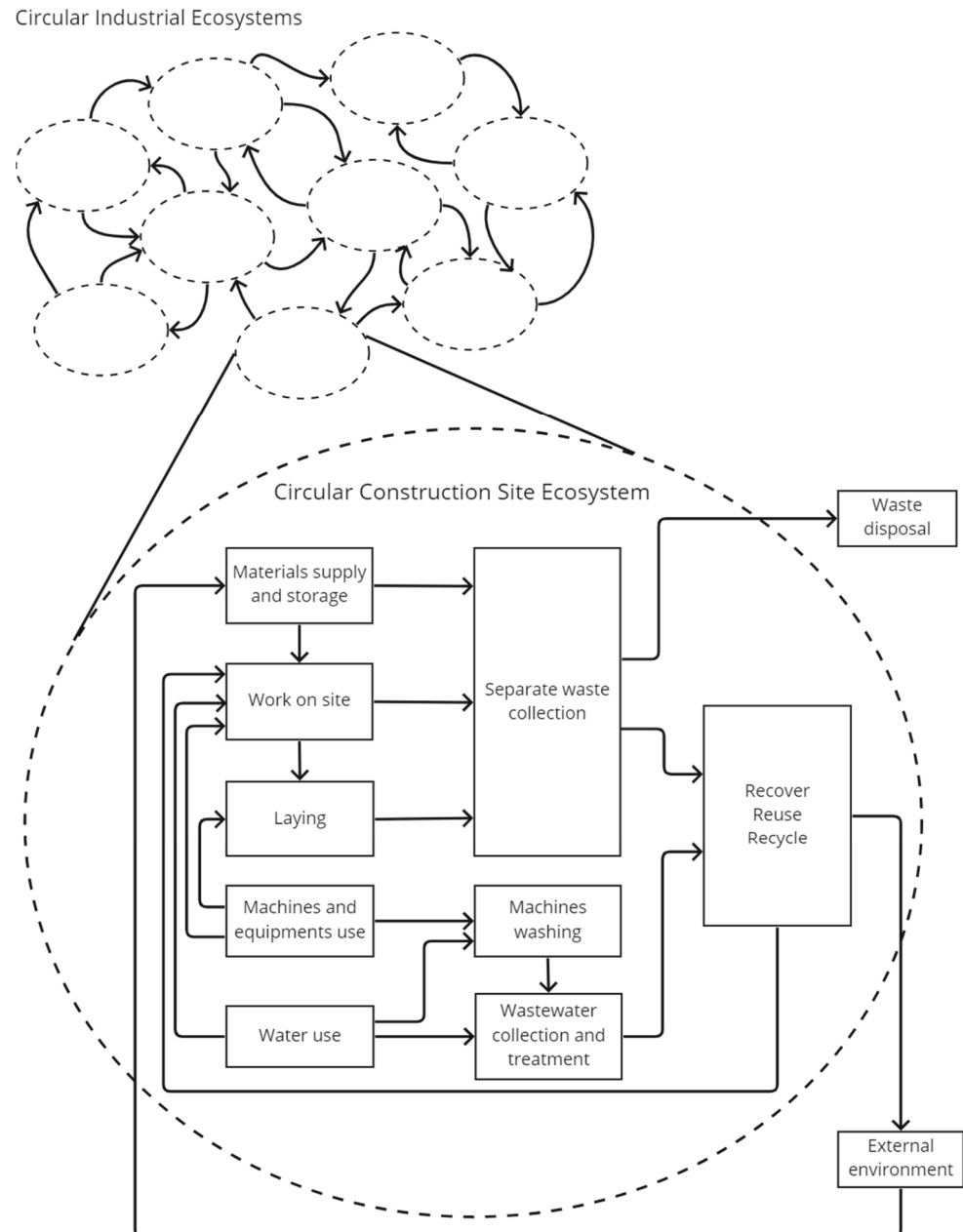

**Figure 8.** Circular construction site ecosystem.

The process is divided into steps because in each of these phases three different types of waste may be produced: packaging, materials, leftovers and scraps of material. They must be sorted and collected in a specific area of the construction site and to avoid being classified as waste a way must be found, where possible, to reuse them.

The sustainable development goal n° 12 "Responsible consumption and production" aims to use sustainable packaging by 2030; several industries have started to develop eco-friendly packaging made of recycled, biodegradable, flexible, and light material. Additionally, construction materials must have sustainable packaging to facilitate their management on-site after the product has been stored.

It could happen that the resources delivered on-site are then processed in a specific working area and leftovers of material are produced. They have to be sorted and stored in the waste management area for future uses. Similar observations can be made when laying finishing elements, such as flooring and tiles; scraps are usually produced from cutting operations of materials. In both of these situations, a reuse and/or recycling strategy can be

planned. We can refer, for example, to internal circularity when from leftover ceramics we can obtain inert materials or aggregates to use in the same construction site and to external circularity when metal scraps are sent to a steel industry to manufacture new steel.

Usually, if there are products that are not used, they can be stored in the site and then used as stock material for maintenance operations during the use phase, or they can be sent to other sites or industries, paying attention to the traceability of materials. Circular strategy may be also be applied to wastewater coming from processes or from the machines' washing area; water must be collected and treated before its return to the environment or before it is reused for other non-potable purposes in the same construction site.

Once the unused or non-usable resources are collected, to efficiently plan their future destination, it is possible to predispose a "No-Waste Management Plan" where non-hazardous materials are selected in order to be directly reused on-site as by-products or to be recovered and recycled outside in industries.

This is complementary to the "Waste Management Plan"; the main difference is that the former considers products and leftovers or scrap materials before they are classified as waste and therefore they do not have to undergo the end-of-waste process to be reused. It also outlines strategies for a possible valorization of these resources while avoiding disposal in landfills.

The rationale behind this methodology is the waste amount reduction and does not involve waste production, or at least involves only a minimum quantity of it. This is possible because more sectors are involved and this allows waste coming from a side to be considered as a resource for other sides.

Therefore, the CCS methodology can be represented as a double cycle, one internal and the other external, where resources, scraps, or waste that come from the construction phase are managed in order to be revalued (Figure 9).

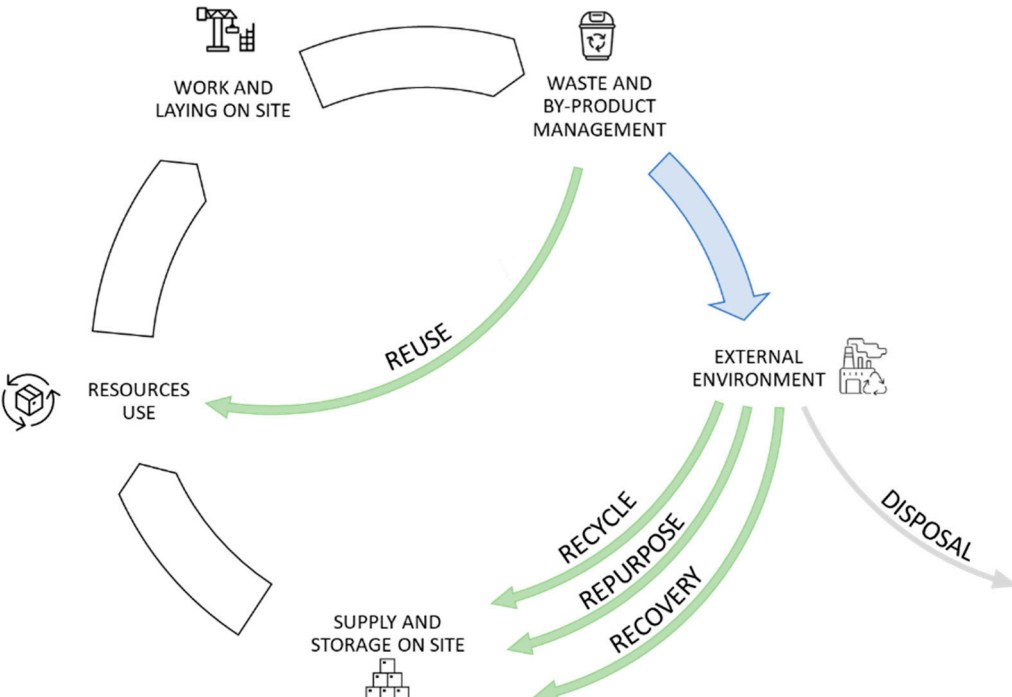

**Figure 9.** Circular construction site (CCS) methodology.

The internal cycle only considers the reuse of material as by-products that do not need special processes, or waste that undergoes downcycling where the new product has a lower value than the original one; instead, the external one considers all the possible circular solutions that can be adopted in order to reintroduce them as resources in new cycles at the same value, with higher value, or with a new purpose.

## 5. KPI Indicators

In order to understand and improve the CCS methodology, it is fundamental to use indicators that give quantitative feedback. The voluntary environmental protocols (LEED; BREEAM; etc.) use several indicators to award buildings but they consider the entire life cycle and only few parameters take into account the choices made during the construction phase. To establish a list of indicators, the ones that could better manage the circularity of the construction stage, in terms of site quality, materials, water, energy, and waste, are selected. After a deep analysis of these protocols, the indicators that refer to the construction stage and that could be useful to the CCS methodology are listed below.

### 5.1. Site Quality

The BREEAM protocol [36] presents the issue "Site selection" to encourage the use of previously occupied or contaminated land; it is split into two parts:

### 5.1.1. Previously Occupied Land

A percentage of the proposed development's footprint is on an area of land which had previously been occupied by industrial, commercial, or domestic buildings, or fixed surface infrastructure. To award credits, the percentage should be at least 75% of the proposed development's footprint on previously developed land.

### 5.1.2. Contaminated Land

According to the ITACA protocol [37], the "level of previous use of the site" can be calculated as follows:

$$\text{Indicator} = Bi/A \times (-1) + Bii/A \times (0) + Biii/A \times (3) + Biv/A \times (5) \text{ [-]}$$

where:
A: site area [m$^2$];
Bi: total area of the site area with soil characteristics in their natural state [m$^2$];
Bii: total area of the site area with green areas and/or on which there were agricultural activities [m$^2$];
Biii: total area of the site area on which there were building structures or infra-structures [m$^2$];
Biv: total surface area of the site area on which remediation operations were conducted (or are planned) [m$^2$].

### 5.2. Material and Resource

Almost all of the protocols present a section related to materials and resources; they give points related to the type of product chosen and its impact on the environment.

In the LEED protocol [35], there are credits about "Building product—disclosure and optimization" to encourage the choice of product with life cycle information available. The building acquires points if a certain percentage of the materials have the following options:

- Environmental product declarations (EPD);
- Multi-attribute optimization;
- Raw material source and extraction reporting;
- Leadership extraction practices;
- Material ingredient reporting;
- Material ingredient optimization;
- Product manufacturer supply chain optimization.

The BREEAM protocol [36] encourages the use of tools to evaluate the life cycle impact (LCI) of construction materials and the selection of materials or products with EPDs; it presents a credit about the "Responsible sourcing of construction products" and also awards points if a "Sustainable procurement plan" is provided.

The category "Eco-friendly materials" of the ITACA protocol [37] presents a list of characteristics that awards the building according to indicators that are calculated as follows:

5.2.1. Reuse of the Existing Structures

It presents an indicator to calculate the percentage of the envelope and ceiling surfaces of the existing building that is reused in the project.

$$\text{Indicator} = Sr_{tot}/S_{tot} \times 100 \ [\%]$$

where:

$Sr_{tot}$: total surface area of the building envelope elements and intermediate floor slabs of the existing building that will be retained and reused in the project [m$^2$];
$S_{tot}$: total surface area of the building envelope elements and intermediate floor slabs before renovation [m$^2$].

5.2.2. Recovered/Recycled Materials

It presents an indicator to calculate the percentage by weight of recycled and/or recovered materials used in the intervention in addition to the legal percentage limit.

$$\text{Indicator} = Pr_{extra}/P_{tot} \times 100 \ [\%]$$

where:

$Pr_{extra}$: the weight of recycled and/or recovered materials used in the building in addition to the legally required minimum quantity [kg];
$P_{tot}$: total weight of materials used for the building [kg].

5.2.3. Material from Renewable Sources

It presents an indicator to evaluate the percentage by weight of renewable source materials used in the work.

$$\text{Indicator} = Pr_{tot}/P_{tot} \times 100 \ [\%]$$

where:

$Pr_{tot}$: the total weight of renewable source materials used in the building [kg];
$P_{tot}$: total weight of materials used in the building [kg].

5.2.4. Local Material

It presents an indicator to establish the weight percentage of local materials compared to those used in the construction.

$$\text{Indicator} = MI/M \times 100 \ [\%]$$

where:

MI: the total weight of locally produced materials/components used for the envelope elements, intermediate floor slabs, and the elevation structure [kg];
M: the total weight of the envelope elements, intermediate floor slabs, and the elevation structure planned in the project [kg].

5.2.5. Disassembly Material

It presents an indicator to calculate the weight percentage of disassembled materials compared to those used in the construction.

$$\text{Indicator} = Pr_{tot}/P_{tot} \times 100 \ [\%]$$

where:

Prtot: the weight of disassembled materials that can be recycled or reused in the project [kg];
Ptot: the total weight of the materials used in the project under consideration [kg].

### 5.2.6. Certified Material

It allows the calculation of the number of products with Type I and Type III environmental marks/declarations.

$$N_{tot} = A \times 1.5 + B \times 0.5 + C \times 1.25 + D \times 1 + E \times 0.5 + F \times 0.5 \text{ [-]}$$

where:

A: total number of products with a Type I label/declaration, according to UNI EN ISO 14024 [53];
B: total number of products with a category EPD, compliant with UNI EN 15804 [34];
C: total number of products with a product-specific EPD, compliant with UNI EN 15804 [34];
D: total number of products with a Type III mark/declaration conforming to UNI EN ISO 14025 [54];
E: total number of products endowed with another environmental mark approved by the ITACA Protocol Promoting Committee;
F: total number of products with a Type II environmental self-declaration in compliance with the UNI EN ISO 14021 standard [55], verified by a conformity assessment body.

### 5.3. Water

The protocols present credits for water management in the building for internal and external uses but this results in them being difficult to adopt in the construction stage because they are focused on potable water use during building operation and the maintenance phase. Some consideration is given to rainwater management, but no indicators are provided.

In this case, it is possible to calculate a general indicator that provides the percentage of water circularity as the "Circular Transition Indicators" (CTIs) proposed by the "World Business Council of Sustainable Development" (WBCSD) for water circularity [56].

$$\% \text{ Water circularity} = (\% \text{ circular water inflow} + \% \text{ circular water outflow})/2 \text{ [\%]}$$

where:

$$\% \text{ Circular inflow total} = (Q \text{ total circular water withdrawal})/(Q \text{ total water withdrawal}) \times 100\% \text{ [\%]}$$

and

$$\% \text{ Circular water outflow} = (Q \text{ total circular water discharge})/(Q \text{ total water withdrawal}) \times 100\% \text{ [\%]}$$

In addition, it also provides an indicator for internal water reuse or recycle.

$$\text{Onsite water circulation} = (Q \text{ water use-Q total water withdrawal})/(Q \text{ total water withdrawal}) + 1 \text{ [-]}$$

### 5.4. Energy

The protocols provide credits for efficient energy use during the building lifecycle and for energy from renewable sources. Similar considerations could be made in the construction site for the energy supply.

The LEED protocol [35] has the credit "Renewable energy production" that gives points on the percentage of renewable energy used in the building per annual energy cost; in this case, the equation should refer to the construction site energy cost.

A more generic indicator is provided by WBCSD [56] to evaluate the percentage of renewable energy on annual consumption:

$$\% \text{ Renewable energy} = (\text{renewable energy})/(\text{total energy}) \times 100\% \text{ [\%]}$$

### 5.5. Waste

The waste management is closely linked to the choice of materials during the construction phase. The protocols mainly present credits to manage waste during the use phase of a construction. The LEED protocol [35] awards buildings that present a C&D waste management plan; the credit "Construction and Demolition waste management" presents two options:

- Diversion

Divert at least 50% of the total construction and demolition material; diverted materials must include at least three material streams.

OR

Divert at least 75% of the total construction and demolition material; diverted materials must include at least four material streams.

- Reduction in total waste material

Do not generate more than 2.5 pounds of construction waste per square foot (12.2 kg of waste per square meter) of the building's floor area.

Additionally, the BREEAM protocol [36] presents a credit to assess construction site waste management, the so-called "Construction waste management"; the issue is split into two parts:

- Construction waste reduction
- Diversion of resources from landfill

The ITACA protocol [37] has a criterion related to the waste generated in the operational phase, but it also has a criterion that establishes a way of calculating the "land balance" to promote the reuse of excavated soil on-site instead of landfilling it as solid waste.

$$\text{Indicator} = Vtr_{tot}/Vs_{tot} \times 100 \, [\%]$$

where:

$Vtr_{tot}$: the total volume of waste soil reused on-site [m$^3$];

$Vs_{tot}$: the total volume of excavations planned or carried out [m$^3$].

From the above, it is clear that only the Italian environmental protocol ITACA provides indicators to quantify the choices made during the construction stage; the other protocols only give threshold values that have to be reached to get points.

The Circular Transition Indicators (CTIs) created by the "World Business Council of Sustainable Development" (WBCSD) were also created to improve the CE model in the companies. Their CTIs could be used in a construction site to monitor its circularity in terms of water and energy as showed above, but it also presents a generic indicator for materials' circularity (Figure 10). It is calculated as the weighted average between the percentage of inflow materials, that can be from renewable or non-virgin contents, and the percentage of outflow materials, determined by the percentage of potential recovery, which is focused on design, and the actual recovery.

$$\% \text{ Material circularity} = (\% \text{ circular inflow total} + \% \text{ circular outflow total})/2 \, [\%]$$

$$\% \text{ Circular outflow total} = [(\%\text{circular outflow A} \times \text{mass A}) + \cdots + (\%\text{circular outflow X} \times \text{mass X})]/(\text{Total mass A} + \cdots + X)$$

These indicators give a comprehensive quantitative idea of the percentage of sustainable resources entering the construction site and the actual percentage of materials reused in the same site (internal circularity). The percentage of outgoing circularity should be analyzed in detail to understand which part undergoes a recovery, recycling, or repurposing process and which is disposed as waste.

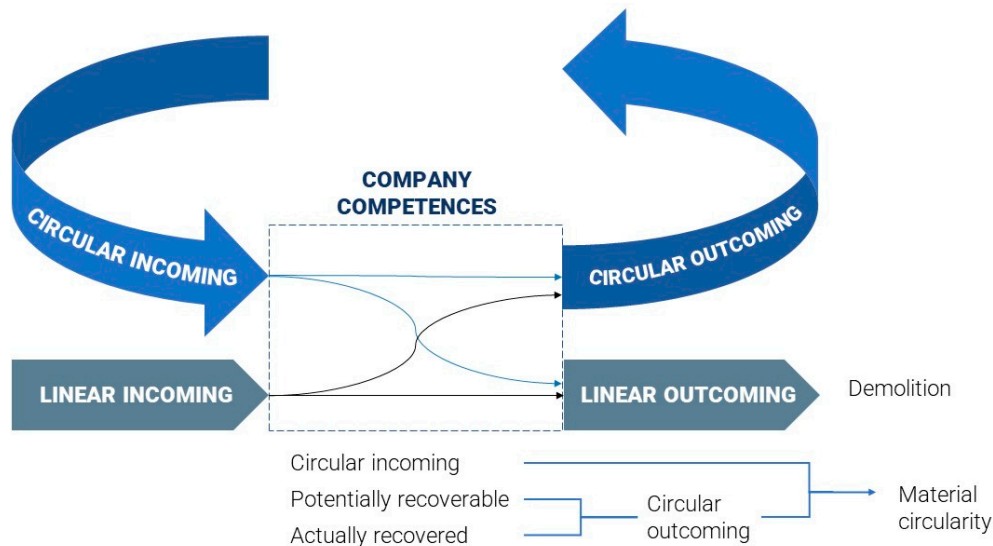

**Figure 10.** Material flow.

## 6. Conclusions

This paper has focused on the development of a methodology to regulate circularity in the construction phase. Based on the circular economy model, the flow that all resources entering a construction site must follow, in order to be revalued and not discarded, was outlined. The aim of this work was not only to promote a new circular system in the AEC industry, but also to make people realize that products may end up having a specific purpose; the resources they are made of should always be recovered and reused for other purposes. Whether this purpose degrades the initial value, or improves it, or remains unchanged, the resources will continue to be used rather than thrown away. This is the goal of the CE model, where cyclicity means that materials are re-used several times into a CCS or into more CCSs and no waste is produced. For this to work better, it is important that all industrial ecosystems are interconnected and, as in a closed loop, exchange resources with each other.

Today, this solution is not often adopted because the different industrial ecosystems do not effectively and efficiently communicate with each other. To enable an easy exchange of information, specific tools are needed, such as the virtual material passport that keeps track of all product information, the no-waste management plan that can properly define its future destination, or the IT platform that can favor exchanges of the data regarding CCSs' material availability and characteristics.

All of these observations may seem to be made only from an environmental point of view, paying attention to the impacts of the construction site on the environment, but it is important to emphasize that CE concepts also include economic and social aspects. Sometimes, the construction industry may feel that these types of choices are not (time-) cost-effective, which is why the European Union and individual countries, such as Italy, are moving towards regulations that promote the principles of the circular economy, incentivizing recycling and energy recovery and taxing landfilling.

In conclusion, the main result of this paper is that the circular construction site (CCS) methodology should be tested on different case studies in order to have a general model that must be enriched according to the type of construction site and by specific information that changes on a case-by-case basis.

Circular choices are more efficient the earlier they occur, particularly at the design stage, as evidenced in the scoring system of the sustainable protocols for certification of constructions (e.g., BREEAM, LEED).

However, "circular" planning of construction activities contributes in a large way to reducing the draw-down from the biosphere.

This paper, in fact, shows how individual construction site ecosystems can find circular economy solutions that close internally (e.g., use excavated soil and rock as permanent materials for works) or externally, both inbound (e.g., use of materials with recycled content) and outbound (e.g., use of soil and rock for off-site nourishment), addressing the issues in the four macro areas that have been explored.

It follows that circular economy solutions become more feasible and effective the more the economic system is oriented toward the adoption of circular solutions, in other words, the more the interconnection between the various worksite ecosystems is enhanced. Therefore, it is advisable that there should be a commitment to the development, provision, and optimization of a network of communicating ecosystems that is parallel and overlapping with the biosphere itself, so as to ensure the flow of matter while preserving the biosphere.

**Author Contributions:** Conceptualization, V.V. and C.F.; methodology, V.V. and M.S.; validation, V.V.; formal analysis, C.F.; investigation, V.V. and C.F.; resources, V.V., M.S. and C.F.; data curation, C.F.; writing—original draft preparation, V.V. and C.F.; writing—review and editing, C.F., M.S. and V.V.; visualization, C.F. and M.S.; supervision, V.V.; project administration, V.V.; funding acquisition, V.V. All authors have read and agreed to the published version of the manuscript.

**Funding:** This research received no external funding.

**Conflicts of Interest:** The authors declare no conflict of interest.

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
