# Peer review of "Architecture Engineering and Construction Industrial Framework for Circular Economy: Development of a Circular Construction Site Methodology"

_sustainability, doi:10.3390/su15031813_

Round 1

Reviewer 1 Report

1.       In Introduction, there are several short paragraphs. Please join the paragraphs with closely related ideas.

2.       Put the reference on the sentence in Lines 30-33.

3.       Lines 83-85: Please complete the sentence. “…with extensive theoretical approaches and practical tools that can change the dynamics, political, social and economic …?”

4.       Lines 86-88: “European Union reports…” Please cite the said report.

5.       Lines 89-92: Make these sentences into 1 sentence. Preferably mentioning the authors and year of instead of [26,27].

6.       Lines 117-120: The author is presenting the proposed methodology – I believe it is their methodology, so no need to cite this statement.

7.       Lines 121-136: Make it one paragraph.

8.       Lines 138: Cite the reference of this statement.

9.       Table 1: Change the title to “Comparison of different environmental protocols in construction industry”. First column, make it “Release Year”. First Row – Change French into France. Put the reference of each environmental protocols included in the table.

10.   Line 221: What flowchart does the author is referring to?

11.   Line 322: Water is primary good as its presence…

12.   Line 403: Specify the type of waste. I suggest “Construction Waste”.

13.   Line 517: The second term of the equation is multiplied by 0, so it is equal to zero – no need to include in the equation.

14.   Lines 519-526: Rewrite it into paragraph form. Also, state the corresponding meaning of the resulting value in the “Indicator”. Whether it is a range of numerical values or single value that would show the “level of previous use of 514 the site”.

15.   Lines 552-556: Rewrite it in paragraph form. Include the corresponding meaning of the resulting percentage.

16.   Follow the above comment and apply to all equations presented in Lines 557-605.

17.   Lines 658-660, write in paragraph form.

Reviewer 2 Report

The manuscript entitled “AEC Industrial Framework for Circular Economy: Development for a Circular Construction Site Methodology” has an interesting and up-to-date topic. Although I believe it can contribute to the literature in the domain, I have some comments that are required to be addressed by the authors before I can suggest it for publication. Please find my comments as follows:

1.      What is AEC in the title? please use its full form in the title. also, I could not find the full form of this abbreviation anywhere in the whole manuscript.

2.      The first line of the abstract saying “All sectors have become disrupted by digital, economic, and demographic transitions” contains a strong claim. Although these transitions have caused challenges and disruptions to some sectors, I believe it is not correct to only highlight the disruptions they had and ignore the benefits they have brought. So, I suggest being more cautious about the presentation of the information especially in the abstract.

3.      Line 16 in the abstract saying “there is no consistent methodology for creating circular and sustainable construction stage at 16 the moment” needs to be more clearly written.

4.      In line 22 in the abstract, what “the circular economy model” refers to? Is it the presented model or a general model available in the literature?

5.      The introduction section is too long, with several pieces of unnecessary information that can be removed.

6.      In line 124, the term “possible methodology” should be replaced with “potential methodology”.

7.      In line 128, “serious” should be “series”.

8.      The manuscript requires to be carefully proofread.

9.      The last paragraph of the introduction is explaining the methodology and steps taken in this research. So, it should be moved to the methodology section. Instead, a paragraph should be added to explain the structure of the paper.

10.   Please increase the size and resolution of Figures 3 and 4, as their texts are not readable.

11.   The manuscript contains too much explanation that can be reduced without decreasing the quality of the main content.

12.   The manuscript lacks a discussion session. I expected to see the results of implementing this methodology in a case study, the validation of the presented instructions, and also a comparison between the presented methodology with the other available methods in the literature in a clear, separate section before the conclusion.

13.   The flow of the manuscript is not proper. There are similar headings in sections 3 and 5. I was lost in the manuscript. so, please re-organize the flow of information in the manuscript. adding a simple figure in the methodology section to present the main steps taken in this research may help a bit.

14.   I believe the literature review should be extended.

Reviewer 3 Report

The topic of the research sounds interesting and covers a timely concern in the construction industry toward a circular economy transition. I believe it can effectively contribute to the circular economy debate. I can recommend it for publication after a minor revision.

1- No need to provide abbreviations in the abstract in case they have not been repeated, such as CCS, and KPIs. In addition, please check the abbreviations in the whole manuscript. For instance, AEC (first time in line 101) has never been introduced in the text (I mean AEC, not AECO).

2- The introduction section needs to be more focused on the main topic of the research, providing a clear definition of the research problem, objective, and contribution of the paper. For instance, the first paragraph (lines 30-36) and the third paragraph (lines 37-44) of this section can be completely removed. Instead, try to provide more detailed content on the research focus. Moreover, the paragraphs in this section are too short (several paragraphs with 2/3 lines) which does not seem professional/academic. You can merge some similar ones to shape it better.

3- in the last two paragraphs of the introduction section please address the associated sections (clearly number them) clearly. In this way, it would be clear what is going to be presented in any section.

4- Figs 3 and 4 are not clear enough and hard to read. If possible, please try to improve the resolution.

5- I was wondering why headings 3 and 4 are the same. Please check, clarify, and do the needful.

6- I could not get how the authors have validated their developed methodology. Based on what you have tested and validated the tool? Please clearly clarify.

7- There are many typos and grammatical errors within the manuscript. Please do professional proofreading before submitting the revised version.

Round 2

Reviewer 2 Report

I see the manuscript has much improved due to the effort authors have put into revising the manuscript based on the comments. I recommend the manuscript in its current form for publication.